# Introducing a Novel Paper Point Method for Isolated Apical Sampling—The Controlled Apical Sampling Device: A Methodological Study

**DOI:** 10.3390/biomedicines13061477

**Published:** 2025-06-15

**Authors:** Christoph Matthias Schoppmeier, Gustav Leo Classen, Silvia Contini, Paul Rebmann, David Brendlen, Michael Jochen Wicht, Anna Greta Barbe

**Affiliations:** 1Polyclinic for Operative Dentistry and Periodontology, Faculty of Medicine and University Hospital Cologne, University of Cologne, 50923 Cologne, Germany; gustav.classen1@uk-koeln.de (G.L.C.); michael.wicht@uk-koeln.de (M.J.W.); greta.barbe@uk-koeln.de (A.G.B.); 2Produits Dentaires SA, 1800 Vevey, Switzerland; silvia.contini@pd-dental.com (S.C.); paul.rebmann@pd-dental.com (P.R.); david.brendlen@pd-dental.com (D.B.)

**Keywords:** 3D-printed tooth model, apical periodontitis, endodontic microbiology, microbial sampling, root canal microbiome

## Abstract

**Objectives**: To introduce a novel method for apical lesion sampling using a protected paper point device and to evaluate its effectiveness and robustness during the sampling process in vitro. **Methods**: A prototype for apical sample collection was developed as an adaptation of the Micro-Apical Placement System—the device features a highly tapered screw head with a thin, hollow, stainless-steel tube and an internal wire piston. Standardized 5 mm paper points (ISO 10; PD Dental, Switzerland) served as carrier material. The prototype was tested using 30 × 3D-printed, single-rooted tooth models inoculated using two bacterial strains (*Staphylococcus epidermidis* and *Escherichia coli*) to simulate apical and intraradicular bacterial infections, respectively. The sampling process involved collecting and analyzing samples at specific timepoints, focusing on the presence or absence of *E. coli* contamination. Following sample collection, cultural detection of bacterial presence was performed by incubating the samples on agar plates to confirm the presence of *E. coli*. Samples were collected as follows: S0 (sterility control of the prototype), P0 (sterility control of the tooth model), P1 (apical sample collected with the CAPS (controlled apical sampling) device, and P2 (contamination control sample to check for the presence of *E. coli* inside the root canal). **Results**: Handling of the CAPS prototype was straightforward and reproducible. No loss of paper points or complications were observed during sample collection. All sterility samples (P0, S0) were negative for tested microorganisms, confirming the sterility of the setup. P2 samples confirmed the presence of *E. coli* in the root canal in all trials. The P1 samples were free from contamination in 86.67% of trials. **Conclusions**: The CAPS method for apical sampling demonstrated advances in the successful and precise sample collection of apically located *S. epidermidis* and will be a useful tool for endodontic microbiological analysis. Its user-friendly design and consistent performance highlight its potential for clinical application, contributing to more accurate microbial diagnostics and later patient-specific therapeutic approaches in endodontic treatments.

## 1. Introduction

Apical periodontitis is an inflammatory disease resulting from microbial colonization of the root canal system, typically occurring in teeth with necrotic pulp or after tooth removal [1,2,3]. Carious lesions produced by bacterial infections are the primary cause [4], with other contributing factors including trauma, periodontal diseases, iatrogenic procedures, and developmental anomalies [5]. Endodontic microbiology has evolved significantly due to technological advances in recent years [4,6]. Initially, open culture methods were employed, providing fundamental microbiological insights [3]. Currently, next-generation sequencing enables comprehensive profiling of bacterial communities within the root canal system. This method has confirmed previous culture findings and established the concept of the microbial community as a pathogenic entity in apical periodontitis [7]. Specifically, 16S rRNA gene sequencing has enhanced understanding of the disease, enabling the precise identification and quantification of microbial species [8].

Generally, the primary endodontic microbiome in the root canal is a mixed infection with approximately 10–30 species per root canal, predominantly consisting of obligate and facultative anaerobic bacteria (mostly gram-negative), which have selective advantages in the endodontic system [9,10]. The bacterial load correlates with the size of the periapical lesion and clinical symptoms [9]. Interestingly, the apical microbiome differs from the intraradicular microbiome [9]. Chávez De Paz found that the microbiome in untreated root canals—as opposed to filled root canals with periapical lesions—shifts from gram-negative to gram-positive bacteria [11].

In many cases, secondary infections in root canal-treated teeth are more severe in terms of healing and disease progression and are increasingly observed in older populations. This can be primarily attributed to the compression of morbidity and the higher residual dentition associated with increasing age [12]. Secondary infections typically involve fewer species, usually one to five, dominated by gram-positive bacteria, with Enterococcus faecalis being the predominant pathogen [13]. Teeth with apical periodontitis (both primary and secondary) often exhibit bacterial biofilms in the apical canal system; the prevalence of these biofilms increases with the size of the lesion [2].

Bacteria at the advanced infection front in the apical part of the root canal system are thought to play a central role in the pathogenesis of apical periodontitis. Therefore, precise decoding of the apical microbiome is crucial, as accurate characterization of microbial communities provides essential insights into disease mechanisms and can contribute to the development of more targeted diagnostic and therapeutic approaches. Traditionally, root canal samples are collected using paper points, as this method is simple and easily integrated into clinical practice. However, paper points primarily capture bacterial cells located in the main lumen of the root canal and its immediate surroundings [14,15]. Additionally, sampling via the coronal access cavity does not allow precise determination of the origin of the samples within the root canal, compromising the accuracy of microbiological analyses. To overcome these limitations, alternative sampling methods have been developed. Endodontic files can collect more material but are difficult to use, with no proven superior effectiveness. Aspiration of canal fluid with saline injection is another method, but it similarly lacks evidence of enhanced efficacy [16]. In contrast, cryopulverization offers a more comprehensive analysis by providing representative samples of the entire root canal system. However, the invasiveness of this method limits its application to extracted teeth or post-surgical samples [17,18,19], highlighting the need for a minimally invasive approach that can be routinely used in clinical practice. Overcoming the limitation of contamination of the coronal and middle root canal sections of paper points, while enabling exclusive apical microbiome determination, would be a significant advancement towards standardized microbial examination.

In this methodological study, we aimed to (i) develop a novel sampling technique for apical lesions using protected paper points with the CAPS (controlled apical sampling) device via the root canal in vitro, and (ii) evaluate its effectiveness and reproducibility during the sampling process, specifically focusing on the precision and consistency of the collected samples and the minimization of contamination risk.

## 2. Materials and Methods

### 2.1. Development and Functionality of the Prototype

This manuscript has been written according to the Preferred Reporting Items for Laboratory studies in Endodontology (PRILE) 2021 guidelines (Figure 1). In collaboration with Produits Dentaires (PD Dental, Vevey, Switzerland), the authors engineered the CAPS prototype for apical sample collection. The goal was to facilitate atraumatic, straightforward, and contamination-free sampling of apical granulomas for microbial analysis. The design was based on the existing Micro-Apical Placement (MAP) system, an accurate tool for retro-obturation, perforation repair, and apical plugs. The MAP system features a syringe-like handpiece and flexible cannulas that enable targeted placement of filling materials (e.g., mineral trioxide aggregate) at the root apex.

The new prototype retains the handpiece of the MAP system but incorporates a softer spring to enhance tactile feedback during sampling (Figure 2). The front section of the device—the primary modification—was specifically tailored for apical sampling requirements. The screw head of the prototype is highly tapered and includes a thin, hollow, stainless-steel tube with an outer diameter of 0.5 mm and a wall thickness of 0.03 mm. This hollow tube is secured to the screw head with epoxy resin. Inside the tube, a stainless steel wire functions as a wire piston, which can extend and retract (Figure 3). A standardized 5 mm piece of a paper point (ISO 10.02 [28], PD Dental) serves as the carrier material for the samples. The original ISO 10.02 paper points (standard length) were manually shortened to 5 mm segments under sterile conditions prior to steam sterilization, to fit the CAPS prototype design. This paper point is affixed to the U-shaped tunnel on the wire piston (Figure 4). The paper point is inserted into the holder and pressed to fix it, leaving 2 mm protruding at the tip (Figure 5, Figure 6 and Figure 7). To maintain contamination-free conditions, the combination of screw head, wire piston, and paper point can be steam sterilized at 120 °C for 20 min. Following sterilization, the screw head, including the wire piston and paper point, is reattached to the application syringe.

### 2.2. Creation of the 3D Model

To simulate sampling under standardized laboratory conditions, a modified, single-rooted, 3D-printed tooth model was developed. This model is based on an endodontic practice model and was designed using graphic software (Blender 4.2.1, Blender Foundation, Amsterdam, The Netherlands). An occlusal trepanation opening was created. A round depression (1 mm deep, 2 mm diameter) was also integrated apically to simulate the apical lesion, with a fluid volume of approximately 6–7 µL. As seen in vivo, apical lesions are about 3–4 mm in diameter on average and thus have a higher fluid volume. In our experiment, however, it is preferable to use a smaller volume to allow the functionality to investigate small apical lesions [20]. The tooth had a length of 20 mm. Thirty 3D teeth were produced using a Digital Light Processing (DLP) 3D printer (Voco Solflex 350, VOCO GmbH, Cuxhaven, Germany) with an XY-resolution of approximately 62 µm and a layer height of 50 µm. Transparent splint resin (V-print splint clear, VOCO GmbH, Cuxhaven, Germany) was used, which is suitable for medical modeling purposes and offers sufficient optical clarity for visual inspection during sampling. The resin exhibits a flexural strength of ≥80 MPa, a flexural modulus of ≥2000 MPa, and a Shore D hardness of ≥82, according to manufacturer specifications. Post-production, the models were washed with isopropanol (≥99.8%) and cleaned in an ultrasonic bath for 5 min. Final curing was performed using a polymerization device that emits 4000 flashes (Otoflash G171, ten flashes per second, NK-Optik GmbH, Baierbrunn, Germany)). The 3D teeth were then randomized. The test procedure is shown in Figure 8.

### 2.3. Bacteria

To test the functionality of the prototype, two different bacterial strains were selected for their macroscopic distinctiveness and ease of cultivation. *Staphylococcus epidermidis* (small, white, smooth colonies) represented the apical microbiome, while *Escherichia coli* (large, gray, irregular colonies) simulated the intraradicular microbiome. Optimal sampling should exclusively yield *S. epidermidis*, without contamination from *E. coli* in the root canal.

One day before the in vitro test, the bacterial strains were inoculated on fresh agar plates (sheep blood agar, Thermo Fisher, Dreieich, Germany) using a three-section streak method. These were incubated at 37 °C for at least 16 h (B 6760, Heraeus Holding GmbH, Hanau, Germany) before use the following day. *S. epidermidis* was used in NaCl solution. A colony of *S. epidermidis* was suspended in 1 mL sterile NaCl solution in a centrifuge tube and vortexed for better suspension (Vortex Shaker, 7-2020m neoLab^®^, Heidelberg, Germany). *E. coli* strains were used directly from the agar plate.

### 2.4. Test Procedure

The experiment was conducted under a safety workbench (HS12, Heraeus Holding GmbH, Hanau, Germany). All canal preparation and sampling procedures were performed by a trained operator (GC). At the start of the instrumentation process, a glide path was created from a 10 K file (Dentsply Sirona Deutschland GmbH, Bensheim, Germany). The cervical, middle, and apical thirds of the simulated canal were instrumented to working length (19.5 mm). The root canal was then sequentially prepared to a size of 50.04 using a reciprocating file (Reciproc blue R50, VDW, Munich, Germany). This was performed using a 6:1 reduction contra-angle handpiece (Dentsply Sirona Deutschland GmbH, Bensheim, Germany) driven by a speed- and torque-controlled electric motor (VDW Gold, Germany), operated according to the manufacturer’s instructions using the Reciproc ALL mode. To mimic clinical conditions, each instrumentation step involved a gentle in-and-out pecking motion. Instrument surfaces were cleaned with gauze soaked in 2.5% NaOCl to remove debris; after each segment of the canal was instrumented, the canal was irrigated with 2.5 mL NaOCl. The tooth models were cleaned with 96% ethanol both externally and internally. To further clean the root canal lumen, sterile paper points (ISO 50.05 [29], Reciproc blue R50, VDW, Germany) soaked in 96% ethanol were introduced into the canal lumen with sterilized tweezers and used to swab the lumen for about 10 s. The ethanol was then allowed to evaporate for 10 min. To remove any remaining ethanol from the canal lumen, a dry, sterile ISO50 paper point was used to dry the canal after the 10 min period.

For the in vitro testing, the following samples were collected: an NaCl control (PC), a sterility test of the 3D tooth (P0), a sterility test of the prototype (S0), the apical sample (P1), and a contamination sample to check for the presence of *E. coli* inside the 3D tooth (P2).

At the start of the experiment, nine agar plates (sheep blood agar, Thermo Fisher) per 3D tooth (a total of 270 plates) were prepared and brought to room temperature. These plates were numbered according to a randomization list (1 to 270). Subsequently, 1 mL of sterile NaCl from a 10 mL tube was pipetted into four centrifuge tubes (sterile polypropylene centrifuge tubes, 15 mL), each labeled according to its sample number (1–270).

The overall sampling workflow per 3D tooth and the corresponding agar plate seedings are summarized schematically in Figure 9 to illustrate the procedure applied uniformly to all 30 specimens.

The PC sample was taken from a sterile 10 mL NaCl tube, with 0.5 mL transferred to an agar plate (sheep blood agar, Thermo Fisher) using a pipetting aid (Pipetus, Hirschmann Laboratory Devices) and sterile, single-use pipettes (1 mL serological pipette, individually wrapped; Corning Incorporated/FALCON, New York, NY, USA). A sterile spatula (plating spatula, PS, blue, sterile; Sarstedt AG & Co. KG, Nümbrecht, Germany) was used to spread the liquid on the agar plate, which was then incubated overnight at 37 °C in an incubator (B 6760, Heraeus Holding GmbH, Hanau, Germany).

For the remaining samples (P0, S0, P1, and P2), both an A-sample and a B-sample of the 3D teeth were collected. The A-sample consisted of a paper point spread on an agar plate. The same paper point was then transferred into 1 mL of sterile NaCl solution (B-sample), followed by 10 s of vortexing. Subsequently, 0.5 mL of the resulting solution was pipetted onto an agar plate and spread. Thus, only one paper point per sample was used, first for direct contact plating (A-sample), then for vortexing and subsequent plating (B-sample).

For the P0 sample, a sterile tweezer and sterile ISO50 paper points (Reciproc blue R50, VDW, Germany) were used to swab the canal lumen of the 3D tooth for 10 s. This paper point was then spread on the designated agar plate and transferred to a centrifuge tube. The agar plate was inoculated using a three-section streak method and placed in the incubator (B 6760 incubator, Heraeus Holding GmbH, Hanau, Germany).

The S0 sample was taken using a sterile ISO15 paper point (Anteos Absorbent Paperpoints, VDW, Munich, Germany) to swab the inside of the hollow cannula of the sampling apparatus for 10 s. This paper point was also spread on an agar plate and transferred to a centrifuge tube.

### 2.5. Inoculation

To simulate the biofilm in the root canal, the 3D tooth was inoculated internally with *E. coli* bacteria. A colony of *E. coli* was picked from the agar plate using a sterile ISO50 paper point (Reciproc blue R50, VDW, Germany), which served solely as a transfer tool to apply bacteria to the internal surfaces of the root canal. After the inoculation step, this paper point was discarded. The inoculated tooth was then placed aside for 30 min to allow bacterial adherence and initial biofilm formation. After this period, the small apical reservoir (apical osteolysis) was filled with 6 µL of the *S. epidermidis* solution using a sterile pipette.

### 2.6. Sampling

The P1 sample was collected using the novel CAPS device. The hollow cannula of the apparatus was inserted 19.5 mm into the 3D tooth. Once the cannula reached the apex, the mechanism was activated, allowing the internal paper point to extend beyond the apex and collect the sample (Figure 10 and Figure 11). This position was maintained for 10 s. Subsequently, the mechanism retracted the paper point into the hollow cannula, and the device was removed from the test tooth. The paper point was then detached from the wire with sterile tweezers and dabbed onto an agar plate, and the sample was transferred to a centrifuge tube.

The final sample (P2) was used to detect the contamination bacterium (*E. coli*) in the root canal of the 3D tooth and was collected at the end of the test. A sterile ISO50 paper point (Reciproc blue R50, VDW, Munich, Germany) was used to swab the root canal for 10 s. This paper point was also initially wiped on an agar plate and then placed in a centrifuge tube. Once all samples were collected, the four centrifuge tubes were further processed. The tubes were vortexed to dislodge the bacteria from the paper point into the NaCl solution. Subsequently, 0.5 mL of the suspension was pipetted onto the B-agar plate and spread with a sterile spatula until the plate absorbed the liquid completely. These samples, along with the A-samples, were incubated overnight and analyzed the following day.

The agar plates were initially inspected visually, and the general appearance of the bacteria used on blood agar was recorded. *S. epidermidis* typically appears as small (1–2 mm), white to gray-white, non-hemolytic colonies. In contrast, *E. coli* is identifiable by its larger (up to 4 mm), gray, smooth colonies that also usually lack hemolytic activity [21].

## 3. Results

The handling of the prototype was straightforward and highly reproducible. No losses of paper points or complications were observed during sample collection. Each of the 30 PC samples was negative. The NaCl solution used was therefore negative for contaminating germs in all tests. Multiple steam sterilizations of the screw head did not negatively impact the fine mechanics within the prototype. All sterility samples, P0 and S0, were interpreted as negative for the tested microorganisms, ensuring that none of the 30 trials were excluded due to a negative control sample.

The P2 samples (both A and B) tested positive for *E. coli* in all 30 cases, confirming the presence of the contaminating microorganism in the root canal in each trial.

In our experimental setup, 26 out of 30 trials (86.67%) demonstrated that both P1A and P1B samples were free from intracanal contamination and contained only the apical microorganism (*S. epidermidis*), indicating successful sample collection (Table 1). In one sample (no. 20), only the P1B sample was free from contamination, while the P1A sample was contaminated with *E. coli*. In three additional trials, both P1 samples were contaminated with *E. coli*.

## 4. Discussion

This methodological study describes the development and first application in vitro of the CAPS instrument for the first time. The method was developed for the specific sample collection of the apical region, and the device was tested for contamination-free apical sampling. We were able to demonstrate that the prototype could effectively collect periapical samples and that the sample quantity was sufficient for analysis—the prototype facilitated simple, reproducible, and largely contamination-free sample collection. Furthermore, the prototype was easy to use, with low susceptibility to errors. Overall, we have shown that the CAPS device could achieve contamination-free periapical samples in over 85% of cases.

### 4.1. CAPS Prototype

The fabrication of the prototype was a material challenge, as it was the first time a hollow needle with an internal wire plunger had been produced with a diameter of only 0.5 mm. A smaller diameter is not currently possible due to material limitations. However, the smallest diameter of the original MAP system was reduced from 0.9 mm to almost half. The attachment of the hollow tube presented significant difficulties. Typically, hard soldering is used for such production components, but this method is not ideal with a wall thickness of only 0.03 mm. Consequently, epoxy resin was utilized, which ensured a solid hold even after multiple sterilization cycles. The most significant challenge was attaching a sampling paper tip to the internal wire. The potential solution was to affix the paper tip to the carrier via an adhesive. However, this method has the disadvantage that the application of a minimal quantity of adhesive is challenging and not straightforward for the user. Furthermore, the migration of the adhesive into the paper tip could result in alterations to its absorption properties, thereby compromising its intended use. Using a shrink tube to connect the paper tip to the stainless steel carrier could have been effective. However, procuring such an extra-thin tube was not possible. An alternative solution was the utilization of a U-shaped tunnel, which provided secure and effective fastening of the paper tip. The limitation of the maximum diameter of the hollow tube meant that the root canal preparation had to be performed to size 50.04. In the majority of cases, this is a feasible approach for root canal revisions [22,23]. Even in initial treatments, an average apical diameter of 0.35 mm is observed [24]. In this context, the paper tip would need to advance approximately 3.5–4 mm at D-3 mm to reach the apical osteolysis, when shaped with an ISO 35.06 [30] Taper to −0.5 mm. For even more precise sampling, the diameter should ideally be further reduced to 0.35 mm with future material innovations.

### 4.2. Sampling Process

During the sampling process, the paper tip remained securely attached to its holder. The sampling process, including the extended periapical placement of the paper tip for approximately 10 s to absorb sufficient fluid, took an average of 20–30 s. Each pass, including the insertion of the paper tip and steam sterilization, required approximately 42–45 min (40 min for sterilization, 2–5 min for attaching the paper point to the wire plunger). For clinical use, multiple screw heads with pre-sterilized wire plunger-paper point combinations could be prepared in advance, allowing multiple tests to be conducted during practice hours.

### 4.3. Conventional Sampling Methods

Previous non-invasive techniques have had significant limitations [16]. For example, the conventional paper point method enables the easy and rapid collection of microbial samples during root canal preparation for subsequent analysis; however, it does not accurately capture microorganisms in hard-to-reach areas such as lateral canals, apical ramifications, dentinal tubules, isthmuses, and recesses. This limitation is particularly significant for the intraradicular identification of the total microbiome but is less critical for the diagnosis and treatment of endodontic disease, where the focus is primarily on the apical region and associated apical periodontitis [9]. While conventional paper points can theoretically extend beyond the apex if sufficient apical preparation is performed, they are generally advanced through the entire root canal via the coronal access. This makes it challenging to precisely target and isolate the apical region without contamination from the coronal and middle thirds. In addition, in the presence of apical exudate or moisture, saturation of the paper tip can occur both in conventional and protected systems. However, the CAPS device was designed to minimize contamination risks from both the coronal and middle portions of the canal and to facilitate controlled and reproducible placement of the paper point directly into the apical region via a guided mechanism. This allows more standardized and targeted apical sampling under visual and mechanical control. Nevertheless, it must be emphasized that moisture-related limitations at the paper tip itself remain inherent to any paper point-based approach, and that for effective application of the CAPS device the root canal should not be completely flooded with exudate or irrigant fluid—conditions that are generally not present during root canal retreatments where careful canal management is performed. Similarly, alternative methods (such as using endodontic files or aspirating NaCl) have limited effectiveness and lack precision in localizing microorganisms within the root canal system. Cryopulverization, while offering more comprehensive and reliable detection of bacteria, is highly invasive and thus not suitable for routine endodontic therapy [16]. The aim should be to develop a simple and standardized sampling technique that is non-invasive and capable of determining the individual microbial colonization of the endodontium, thereby enabling patient-specific endodontic therapy in the future. A first step in this direction is the development and testing of a novel apical sampling method, such as our CAPS prototype.

### 4.4. Bacterial Colonization

Although the 3D-printed tooth model does not contain organic tissue, NaOCl irrigation was applied during instrumentation to mimic the clinical preparation workflow, prevent debris accumulation, and ensure a standardized preparation protocol across all test specimens. In our study, *E. coli* and *S. epidermidis* (both risk group II) were utilized to simulate bacterial colonization within the root canal and in apical osteolysis. These bacteria were selected due to their specific characteristics, low malignancy, and ease of identification based on distinct colony morphology [25,26].

While *E. coli* is not typically associated with endodontic infections, its inclusion in our study enabled clear detection of potential intracanal contamination. In contrast, *S. epidermidis* served as a representative organism for endodontic infections due to its ability to form biofilms and cause persistent infections [27]. Our results demonstrated that contamination by *E. coli* could be reliably detected and was clearly distinguishable from *S. epidermidis*. This distinction facilitated the evaluation of sample purity and the assessment of the efficacy of the new CAPS sampling device.

### 4.5. Limitations

Our in vitro study has several limitations. Translation to clinical practice is somewhat restricted due to the use of a 3D tooth model, which cannot fully replicate real oral conditions. Notably, the mechanical properties of the synthetic materials used in the 3D models differ from natural dentin and pulp tissue, which could affect the accuracy of the results. Specifically, the resin used for 3D printing was a transparent splint resin (V-print splint clear), which is relatively elastic and was selected primarily for its optical properties to allow visual control during sampling, not because it mimics the mechanical behavior of dentin. Additional factors such as temperature fluctuations, the presence of dentinal tubules within the resin matrix of the 3D tooth, and saliva contamination are challenging to accurately reproduce in a laboratory setting. A quantitative analysis of the contaminated samples was not feasible due to the varying levels of *E. coli* present (Figure 8 and Figure 10). As a result, only a qualitative assessment of the contamination was possible. Furthermore, long-term effects and material aging cannot be adequately simulated in such an environment, partially limiting the generalizability of our findings. Finally, while the sample size in this pilot study is sufficient for initial testing, it may need to be increased in further studies to ensure statistical robustness and better capture of variability in clinical settings. Future studies should first test the prototype ex vivo on extracted teeth, followed by initial sample collection during routine root canal treatments in a clinical setting. In addition, the applicability of the device to curved root canals remains to be evaluated, as the current design is optimized primarily for straight canals in the present 3D model. The thin-walled stainless-steel cannula exhibits a certain degree of flexibility due to its fine structure, which represents a deliberate compromise between mechanical stability and flexibility. However, the current prototype requires a sufficiently prepared apical diameter (ISO 50.04 in this study) and is therefore not yet suitable for strongly curved or narrow canals without prior appropriate preparation. Future device iterations should aim to further optimize this balance and improve flexibility to enable broader clinical applicability. Potential technical risks include jamming of the paper point if swelling occurs after absorbing periapical fluid, as well as mechanical stress on the thin-walled (0.03 mm) hollow tube. However, it should be noted that during the present in vitro tests, including multiple cycles of steam sterilization, no instances of breakage, jamming, or inability to retract the paper point were observed. Nonetheless, future iterations of the device and clinical studies should further assess its performance and safety in more complex root canal anatomies.

## 5. Conclusions

The CAPS device—a novel prototype for contamination-free apical sampling—shows promise for advancing endodontic microbiological analysis. Its ability to reliably collect sufficient periapical samples with minimal contamination in vitro, combined with its user-friendly design, highlights its potential value for future clinical applications. The innovative features of the CAPS prototype, including a secure sampling mechanism and effective sterilization process, demonstrated consistent and reproducible results under experimental conditions. While these findings represent an encouraging step towards improving the accuracy of microbial sampling in endodontic treatment, further studies—including ex vivo and clinical investigations—are required to validate its clinical efficacy. In the long term, more accurate identification of the pathogenic microorganisms involved may contribute to the development of future patient-specific therapeutic approaches, provided that these initial results can be confirmed in more complex clinical settings.

## Figures and Tables

**Figure 1 biomedicines-13-01477-f001:**
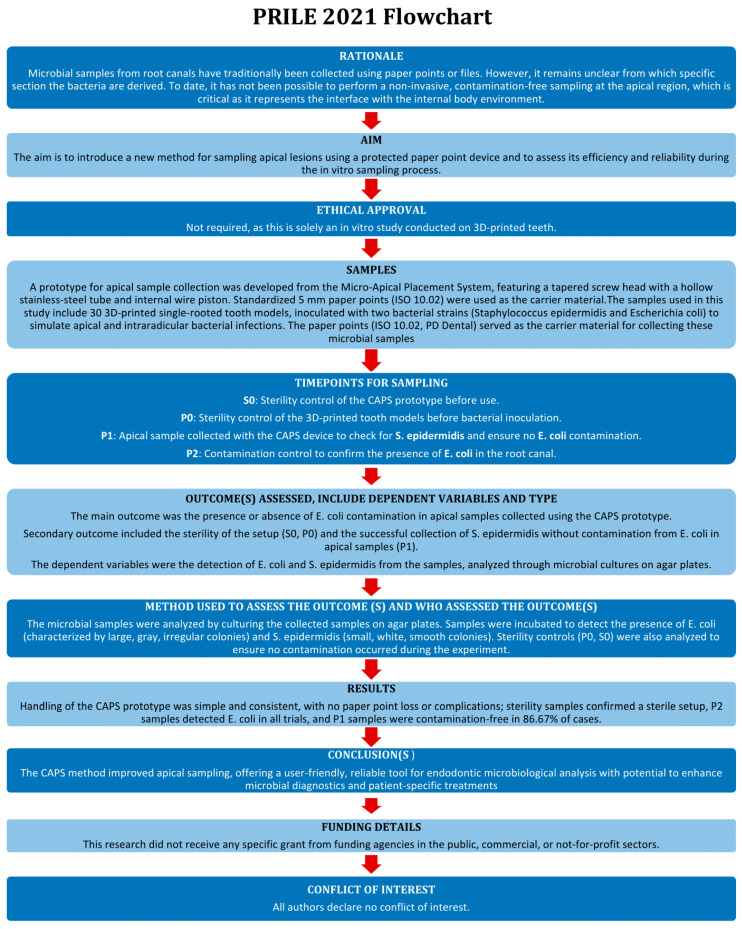
PRILE-2021 flowchart.

**Figure 2 biomedicines-13-01477-f002:**
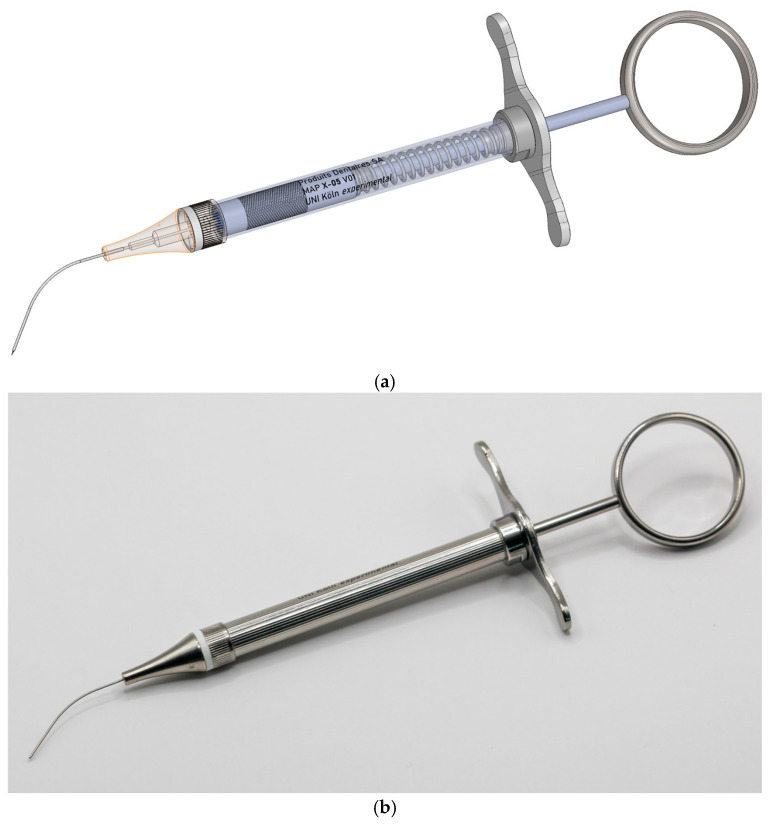
Three-dimensional (3D) controlled apical sampling (CAPS) device with a modified soft spring and customized screw head for apical sample collection: (**a**) transparent and (**b**) original image of the device.

**Figure 3 biomedicines-13-01477-f003:**
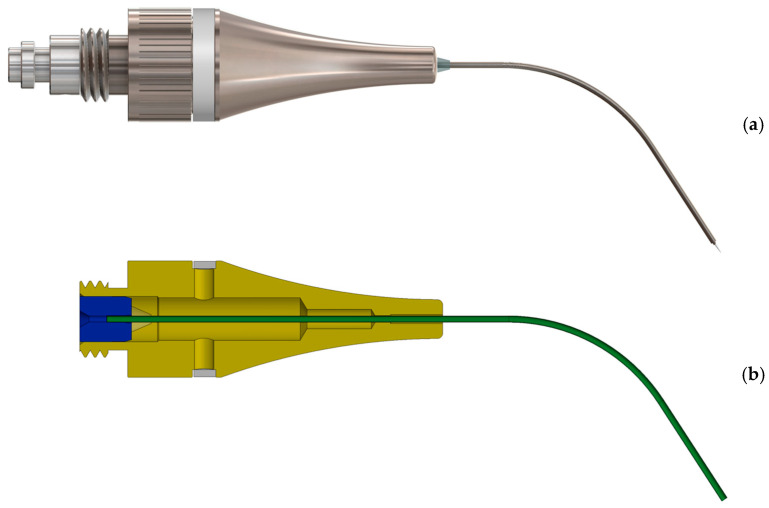
Detailed view of the screw head: (**a**) with hollow tube (outer diameter 0.5 mm) and threading for secure attachment to the syringe; (**b**) transparent to show the attachment of the hollow tube with epoxy resin. Inside the hollow tube, a stainless steel wire functions as a piston.

**Figure 4 biomedicines-13-01477-f004:**
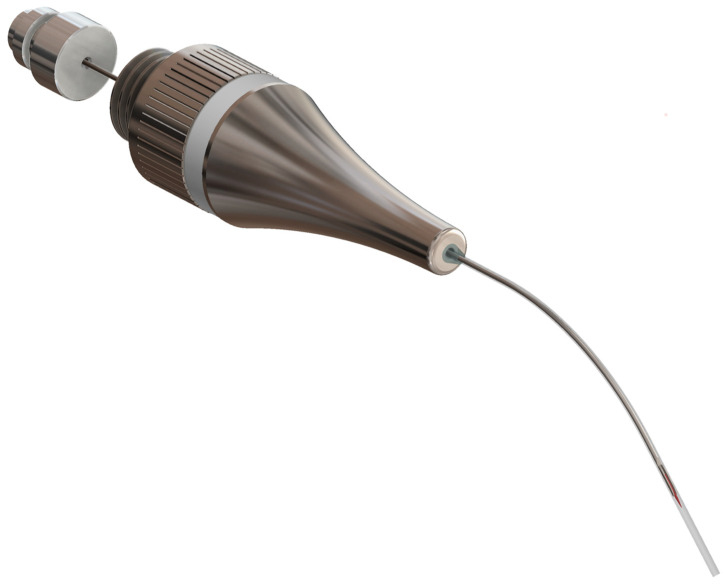
Lateral detailed view of the screw head with its stainless steel wire piston, capable of extending and retracting within the hollow tube.

**Figure 5 biomedicines-13-01477-f005:**
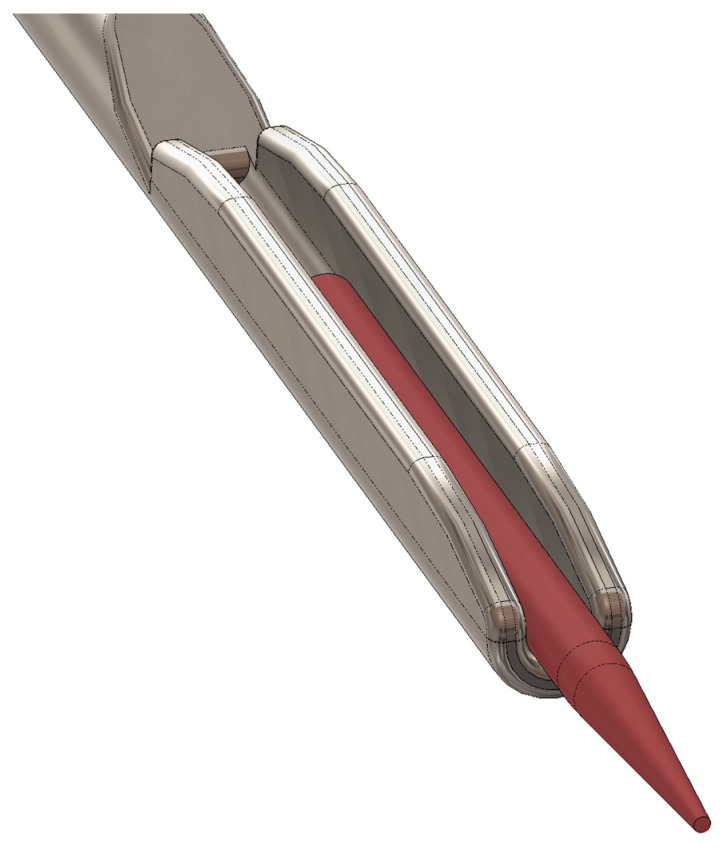
U-shaped end of the stainless steel wire with inserted paper point.

**Figure 6 biomedicines-13-01477-f006:**
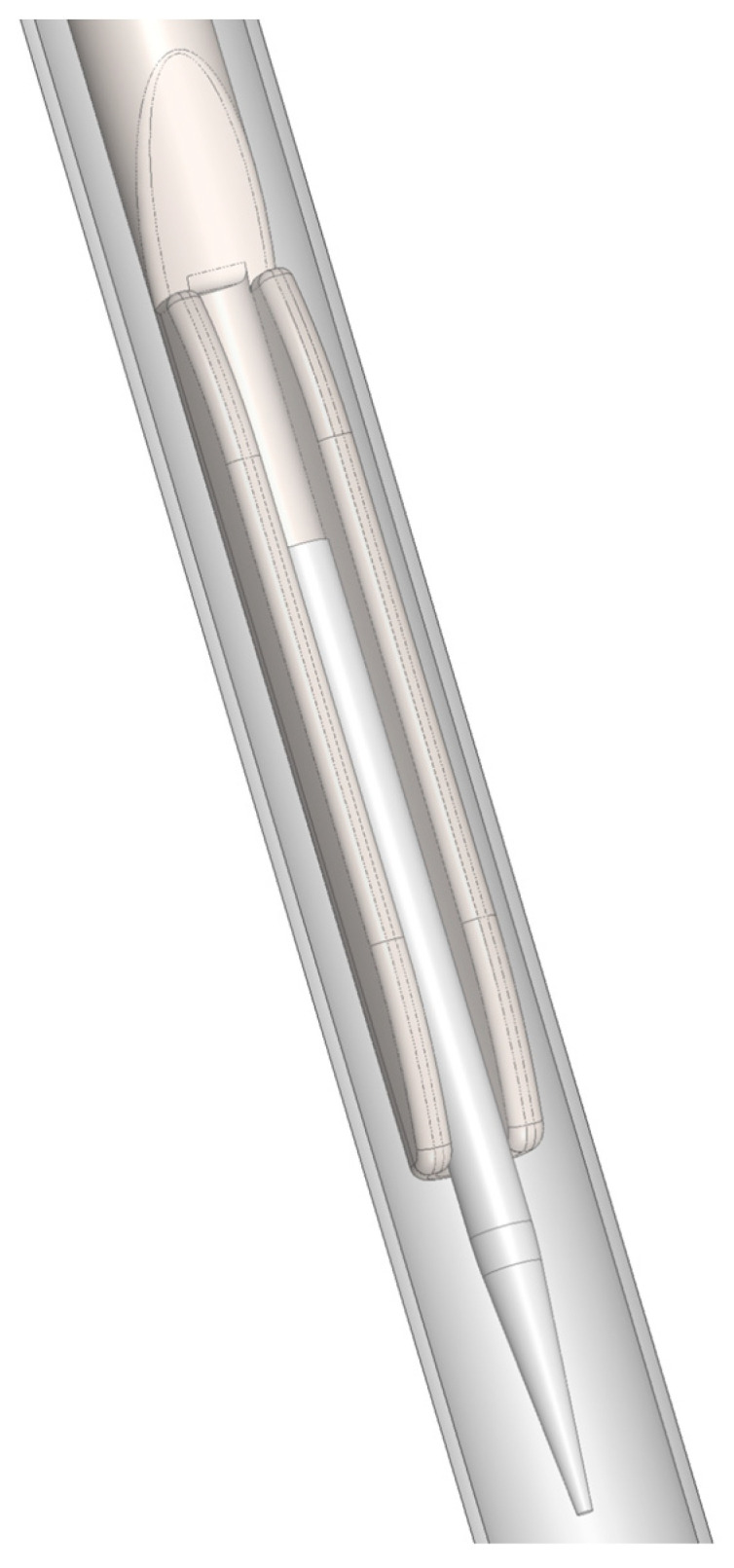
Paper point inserted into the holder, leaving 2 mm protruding from the tip.

**Figure 7 biomedicines-13-01477-f007:**
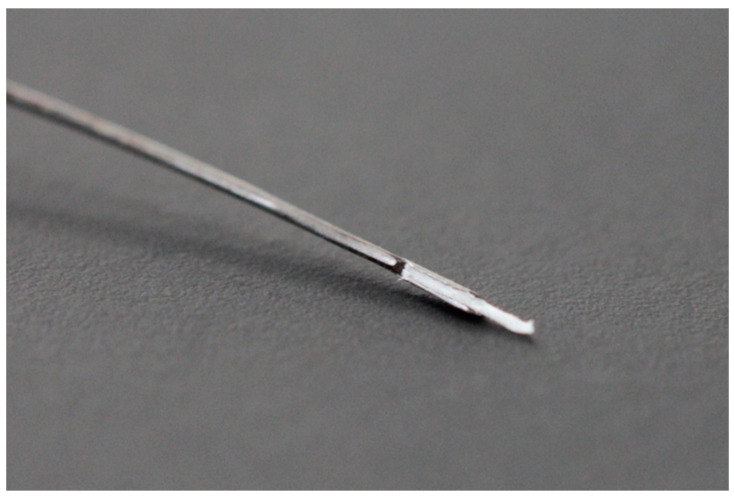
Original image of ISO10 paper point fitted into the U-shaped holder.

**Figure 8 biomedicines-13-01477-f008:**
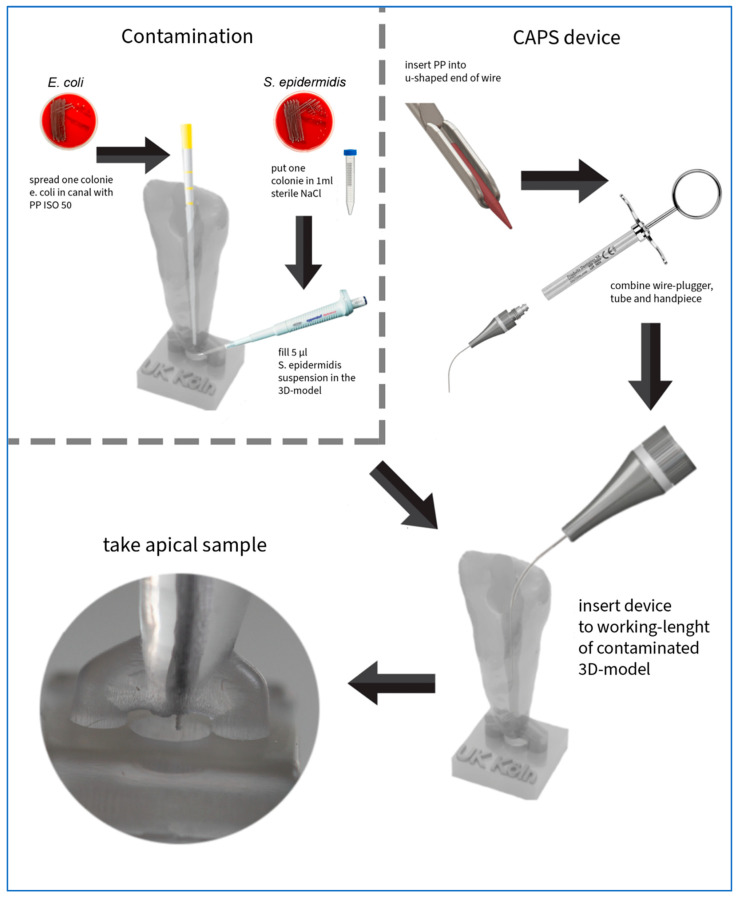
Schematic workflow of sample preparation and collection process using the controlled apical sampling (CAPS) device.

**Figure 9 biomedicines-13-01477-f009:**
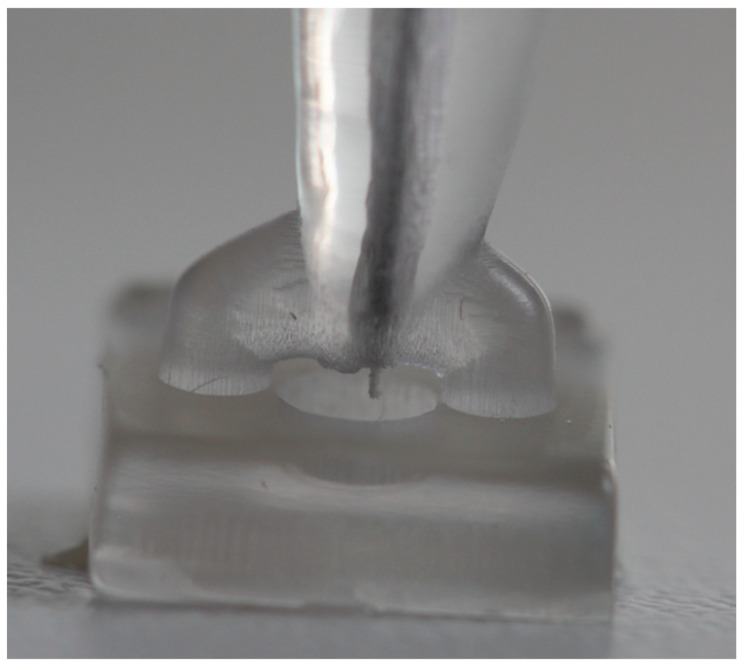
Illustration of sample collection with the visible ISO10 paper point extending beyond the apex to collect bacteria from the simulated apical osteolysis.

**Figure 10 biomedicines-13-01477-f010:**
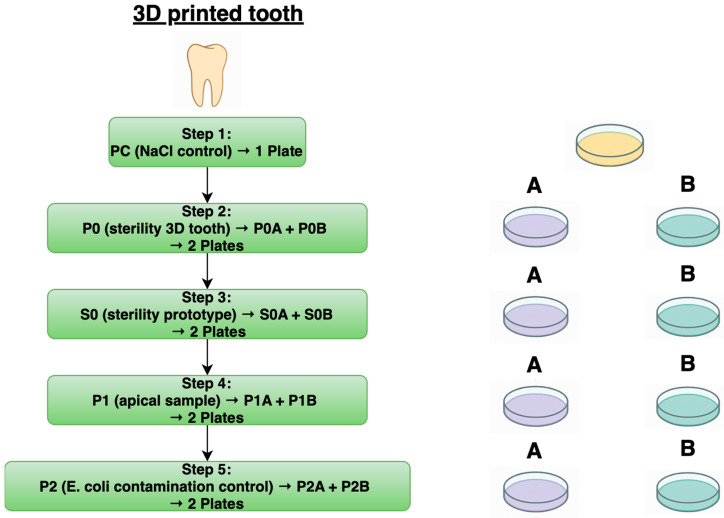
Schematic workflow of the sampling procedure and corresponding agar plate seedings per 3D-printed tooth.

**Figure 11 biomedicines-13-01477-f011:**
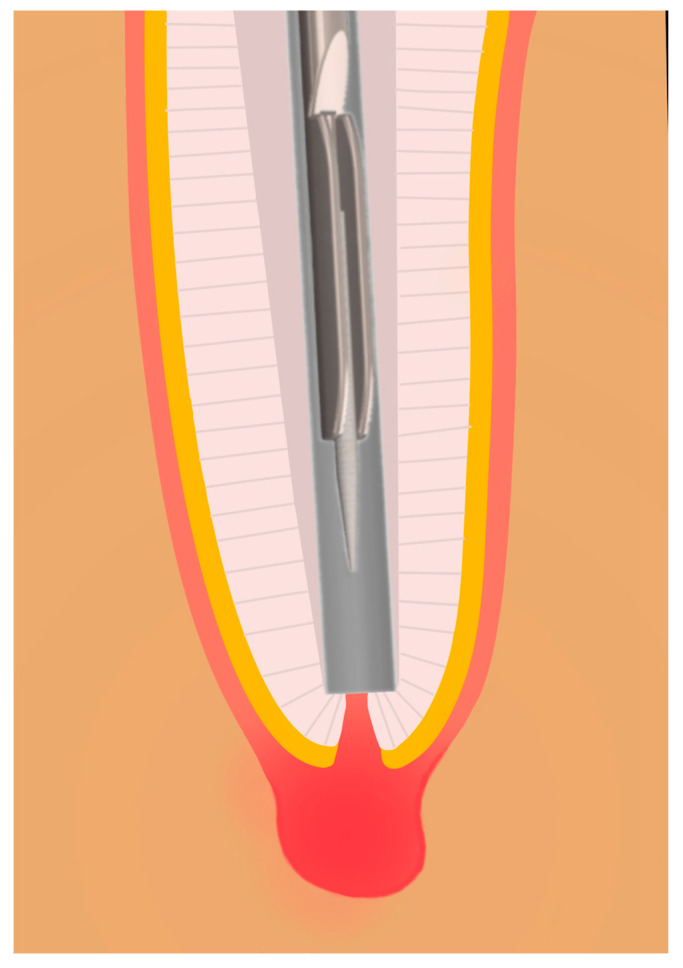
Schematic representation of the sampling method. After mechanical preparation, the hollow tube of the controlled apical sampling (CAPS) device is advanced as close as possible to the apical constriction for sample collection.

**Table 1 biomedicines-13-01477-t001:** Apical sample contamination.

	P1A	P1B
n (%)	30 (100.0)	30 (100.0)
Uncontaminated (%)	26 (86.7)	27 (90.0)
Contaminated (%)	4 (13.3)	3 (10.0)

Abbreviations: P1A, apical sample applied directly to agar plate; P1B, apical sample after vortexing in saline solution and plating.

## Data Availability

The raw data supporting the conclusions of this article will be made available by the authors on request.

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
