# Peer review of "Introducing a Novel Paper Point Method for Isolated Apical Sampling—The Controlled Apical Sampling Device: A Methodological Study"

_biomedicines, 2025, doi:10.3390/biomedicines13061477_

Round 1
Reviewer 1 Report
Comments and Suggestions for Authors
The study presents an adaptation of the Micro-Apical Placement System device (Produits Dentaires, Switzerland), which is commonly recommended for filling retrograde obturations, root perforations, and radicular resorptions, so that the device could be used for sampling microbes from the apical third of the root and the periapical space. The tip for this device was further narrowed and provided with U shaped recipient of the paper point, which is designed to be pushed into the apical space, then retracted back into the tube, after reaching the apex with the tube. After removing from the canal it is foreseen to do microbial evaluation. Study was conducted on 30 transparent resin printed teeth (resin for clear orthodontic splints - Voco Soflexs).
What is ISO 10.02? Paper points usually have the length more than 10mm, and you claim only 5mm, and the taper 2%, i.e. .02. When trying to find out what is ISO 10.2 I found nothing. Did you get that short paper points by cutting the standard length points? Clarify this issue in your text.
Figure 1 needs larger font size. The text in flowchart has drastically smaller font size than the text in the manuscript and even enlarged it is hard to read, especially the white letters with blue background.
How can hollow stainless-steel tube reach the apex of the teeth with possible wide range of curves.
You claim that you introduce an infection with E. coli by paper point 50. Did the paper point reach the apex? What was anyway the size of canal of the 3D printed resin tooth initialy?
Line 220. You didn’t explain A and B –samples meaning. It is not clear in further explanation what is A and B samples! Was each 3D printed teeth tested with 2 paper points, one for direct cultivation on agar plate, and other for dissolution in NaCl and subsequent cultivation?
Line 239. Specify if the “infected” paper point was placed in canal to the working length to provide E. coli inoculation along whole canal length.
Line 342. You claim “Due to the material properties of paper points, the conventional paper point method cannot effectively collect samples beyond the apex, as the paper tip quickly becomes saturated with moisture, preventing the application of apical pressure” is hardly solved with your device, since if there is moieties and exudation in the apex this will happen with your paper point too. Further if the canal is not dry because of exudation the tube holding paper point will be filled with moisture, the same way as paper point alone, and if the canal is dry, the alone paper point will be firm enough to push perplex, especially with such big apical diameter (0.50).
Line 358. You have described the appearance of colonies of both types of bacteria in at least three places. There is no need to repeat.
In the limitation segment you didn’t mention the applicability of the device in curved canals, the present risk of setup breakage since its dimensions are extremely small (0.03 mm wall thickness), jamming and inability to retract the protruding part of the paper point after it swells from absorbing periapical fluid. The problem of differing printed teeth in mechanical properties should be further clarified, and emphasized that the resin you used for teeth printing is pretty elastic, since it is predicted for splints, and that it was used only because it is transparent and not because it mimics dentin properties.
Mitigate the revolutionary nature of the conclusion, especially given that it is a pilot study with a fair number of limitations.
Which was the point of using NaOCl for resin tooth irrigation? There are no organic tissues to be dissolved?
I suggest that you create a diagram that will more clearly present how each of the 30 teeth was tested. Was the same done for all 30 teeth? Present schematically the 9 methodologies you used, i.e. the 9 seedings (agar plates).
Author Response
Comments 1:
What is ISO 10.02? Paper points usually have the length more than 10mm, and you claim only 5mm, and the taper 2%, i.e. .02. When trying to find out what is ISO 10.2 I found nothing. Did you get that short paper points by cutting the standard length points? Clarify this issue in your text.
Replies 1:
The paper points used were ISO 10.02 points (as provided by PD Dental, Switzerland), which indicates ISO size 10 with a 2% taper. These paper points are supplied in standard length. For compatibility with the design of our CAPS prototype, we manually shortened these standard paper points to 5 mm segments under sterile conditions prior to steam sterilization. We have now revised the text accordingly to clearly describe this process and to avoid any potential misunderstanding. We greatly appreciate this opportunity to improve the clarity of our manuscript.
"The original ISO 10.02 paper points (standard length) were manually shortened to 5 mm segments under sterile conditions prior to steam sterilization, to fit the design of the CAPS prototype."
Comments 2:
Figure 1 needs larger font size. The text in flowchart has drastically smaller font size than the text in the manuscript and even enlarged it is hard to read, especially the white letters with blue background.
Replies 2:
The font size in Figure 1 has been adjusted to ensure better readability and to match the manuscript’s overall formatting.
Comments 3:
How can hollow stainless-steel tube reach the apex of the teeth with possible wide range of curves.
Replies 3
Indeed, the current CAPS prototype represents a deliberate compromise between flexibility and mechanical stability, in order to ensure safe handling and avoid breakage during sampling. The thin-walled stainless-steel cannula provides a certain degree of flexibility, but the device requires a sufficiently prepared apical diameter (ISO 50.04 in this study) and is not intended for highly curved or narrow canals at this stage. We have further clarified this limitation in the revised manuscript.
"The thin-walled stainless-steel cannula exhibits a certain degree of flexibility due to its fine structure, which represents a deliberate compromise between mechanical stability and flexibility. However, the current prototype requires a sufficiently prepared apical diameter (ISO 50.04 in this study) and is therefore not yet suitable for strongly curved or narrow canals without prior appropriate preparation. Future device iterations should aim to further optimize this balance and improve flexibility to enable broader clinical applicability."
Comments 4:
You claim that you introduce an infection with E. coli by paper point 50. Did the paper point reach the apex? What was anyway the size of canal of the 3D printed resin tooth initialy?
Replies 4:
The 3D-printed tooth model was designed with an initial apical canal diameter of approximately ISO 15. However, due to typical minor dimensional deviations and potential resin printing artifacts (e.g., polymerization shrinkage), the actual initial apical diameter was expected to be slightly smaller. Therefore, canal scouting was initially performed using an ISO 10 K-file to safely establish the glide path, as also described in the Methods section. The canal was subsequently fully instrumented to ISO 50.04, ensuring standardized preparation throughout the cervical, middle, and apical thirds.
Regarding E. coli inoculation: a sterile ISO50 paper point was used solely as a transfer tool to coat the internal canal walls with E. coli, and it was not inserted beyond the prepared working length (WL - 0.5 mm). No deliberate inoculation of the apical lesion (osteolysis reservoir) with E. coli occurred, which is also reflected by the microbiological results: apical P1 samples consistently demonstrated S. epidermidis as the dominant organism, confirming that no over-inoculation of the apical region with E. coli took place.
Comments 5:
Line 220. You didn’t explain A and B –samples meaning. It is not clear in further explanation what is A and B samples! Was each 3D printed teeth tested with 2 paper points, one for direct cultivation on agar plate, and other for dissolution in NaCl and subsequent cultivation?
Replies 5:
We confirm that for each sample (P0, S0, P1, P2), the same paper point was first applied directly to an agar plate (A-sample) and was then subsequently transferred into 1 mL of sterile NaCl, vortexed, and the resulting suspension plated (B-sample). We have now clarified this procedure in section 2.4 (Test Procedure) for improved transparency.
"For the remaining samples (P0, S0, P1, and P2), both an A-sample and a B-sample of the 3D teeth were collected. The A-sample consisted of a paper point spread on an agar plate. The same paper point was then transferred into 1 mL of sterile NaCl solution (B-sample), followed by 10 s of vortexing. Subsequently, 0.5 mL of the resulting solution was pipetted onto an agar plate and spread. Thus, only one paper point per sample was used, first for direct contact plating (A-sample), then for vortexing and subsequent plating (B-sample).
Comments 6:
Line 239. Specify if the “infected” paper point was placed in canal to the working length to provide E. coli inoculation along whole canal length.
Replies 6:
We acknowledge that the original wording in section 2.5 might have caused some confusion regarding the use of the ISO50 paper point during the inoculation step. We have now revised the section to clarify that the paper point was used solely as a transfer tool to apply E. coli to the internal surfaces of the root canal, and was discarded immediately thereafter. All subsequent sampling was performed with new sterile paper points, as described in the Methods section.
"To simulate the biofilm in the root canal, the 3D tooth was inoculated internally with E. coli bacteria. A colony of E. coliwas picked from the agar plate using a sterile ISO50 paper point (Reciproc blue R50, VDW, Germany), which served solely as a transfer tool to apply bacteria to the internal surfaces of the root canal. After the inoculation step, this paper point was discarded. The inoculated tooth was then placed aside for 30 min to allow bacterial adherence and initial biofilm formation. After this period, the small apical reservoir (apical osteolysis) was filled with 6 µL of the S. epidermidis solution using a sterile pipette."
Comments 7:
Line 342. You claim “Due to the material properties of paper points, the conventional paper point method cannot effectively collect samples beyond the apex, as the paper tip quickly becomes saturated with moisture, preventing the application of apical pressure” is hardly solved with your device, since if there is moieties and exudation in the apex this will happen with your paper point too. Further if the canal is not dry because of exudation the tube holding paper point will be filled with moisture, the same way as paper point alone, and if the canal is dry, the alone paper point will be firm enough to push perplex, especially with such big apical diameter (0.50).
Replies 7:
We fully agree that our device does not fundamentally overcome the limitations related to moisture or exudate at the apical region, as saturation of the paper tip can occur in both conventional and protected systems. We have accordingly revised the section "Conventional Sampling Methods" to clarify this point and to more accurately emphasize that the main advantage of the CAPS device lies in its ability to minimize contamination from both the coronal and middle thirds of the canal and to provide controlled and reproducible apical placement of the paper point. We have also clarified that effective use of the device requires that the root canal is not completely flooded with exudate or irrigant fluid, which is typically not the case in root canal retreatments performed under proper canal management.
"While conventional paper points can theoretically extend beyond the apex if sufficient apical preparation is performed, they are generally advanced through the entire root canal via the coronal access. This makes it challenging to precisely target and isolate the apical region without contamination from the coronal and middle thirds. In addition, in the presence of apical exudate or moisture, saturation of the paper tip can occur both in conventional and protected systems. However, the CAPS device was designed to minimize contamination risks from both the coronal and middle portions of the canal and to facilitate controlled and reproducible placement of the paper point directly into the apical region via a guided mechanism. This allows more standardized and targeted apical sampling under visual and mechanical control. Nevertheless, it must be emphasized that moisture-related limitations at the paper tip itself remain inherent to any paper point-based approach, and that for effective application of the CAPS device the root canal should not be completely flooded with exudate or irrigant fluid—conditions that are generally not present during root canal retreatments where careful canal management is performed."
Comments 8:
Line 358. You have described the appearance of colonies of both types of bacteria in at least three places. There is no need to repeat.
Replies 8:
We fully agree that the description of bacterial colony morphology should not be unnecessarily repeated. We have accordingly revised the section “Bacterial Colonization” in the Discussion to provide a more concise summary and have removed redundant descriptions already provided in the Methods section.
"Although the 3D-printed tooth model does not contain organic tissue, NaOCl irrigation was applied during instrumentation to mimic the clinical preparation workflow, prevent debris accumulation, and ensure a standardized preparation protocol across all test specimens. In our study, E. coli and S. epidermidis (both risk group II) were utilized to simulate bacterial colonization within the root canal and in apical osteolysis. These bacteria were selected due to their specific characteristics, low malignancy, and ease of identification based on distinct colony morphology [24] [25].
While E. coli is not typically associated with endodontic infections, its inclusion in our study enabled clear detection of potential intracanal contamination. In contrast, S. epidermidis served as a representative organism for endodontic infections due to its ability to form biofilms and cause persistent infections [26]. Our results demonstrated that contamination by E. coli could be reliably detected and was clearly distinguishable from S. epidermidis. This distinction facilitated the evaluation of sample purity and the assessment of the efficacy of the new CAPS sampling device."
Comments 9:
In the limitation segment you didn’t mention the applicability of the device in curved canals, the present risk of setup breakage since its dimensions are extremely small (0.03 mm wall thickness), jamming and inability to retract the protruding part of the paper point after it swells from absorbing periapical fluid.
Replies 9:
We have now revised the Limitations section to explicitly address the applicability of the device in curved canals as well as potential technical risks, including breakage of the thin-walled hollow tube, jamming, and difficulties in retracting the paper point after fluid absorption. We have also clarified that no such issues were observed during the present in vitro tests, despite multiple sterilization cycles, but we fully acknowledge that these aspects should be further evaluated in future studies and clinical applications
"In addition, the applicability of the device in curved root canals remains to be evaluated, as the current design is optimized mostly for straight canals in the present 3D model. Potential technical risks include jamming of the paper point if swelling occurs after absorbing periapical fluid, as well as mechanical stress on the thin-walled (0.03 mm) hollow tube. However, it should be noted that during the present in vitro tests—including multiple cycles of steam sterilization—no instances of breakage, jamming, or inability to retract the paper point were observed. Nonetheless, future iterations of the device and clinical studies should further assess its performance and safety in more complex root canal anatomies."
Comments 10:
The problem of differing printed teeth in mechanical properties should be further clarified, and emphasized that the resin you used for teeth printing is pretty elastic, since it is predicted for splints, and that it was used only because it is transparent and not because it mimics dentin properties.
Replies 10:
We have now revised the Limitations section accordingly and clarified that the resin used for 3D printing (V-print splint clear) is relatively elastic, as it is designed for splints, and that it was chosen solely for its transparency to allow visual control—not to mimic dentin properties. This limitation is now explicitly stated to improve transparency of the experimental setup.
"Specifically, the resin used for 3D printing was a transparent splint resin (V-print splint clear), which is relatively elastic and was selected primarily for its optical properties to allow visual control during sampling—not because it mimics the mechanical behavior of dentin."
Comments 11:
Mitigate the revolutionary nature of the conclusion, especially given that it is a pilot study with a fair number of limitations.
Replies 11:
We have revised the Conclusion section to tone down overly ambitious wording and to better reflect the pilot nature of our study. The revised Conclusion now highlights the promising potential of the CAPS device while clearly acknowledging the need for further ex vivo and clinical validation before clinical application.
"5. Conclusions
The CAPS device—a novel prototype for contamination-free apical sampling—shows promising potential for advancing endodontic microbiological analysis. Its ability to reliably collect sufficient periapical samples with minimal contamination in vitro, combined with its user-friendly design, highlights its possible value for future clinical application. The innovative features of the CAPS prototype, including a secure sampling mechanism and effective sterilization process, demonstrated consistent and reproducible results under experimental conditions. While these findings represent an encouraging step towards improving the accuracy of microbial sampling in endodontic treatment, further studies—including ex vivo and clinical investigations—are required to validate its clinical efficacy. In the long term, more accurate identification of the pathogenic microorganisms involved may contribute to the development of future patient-specific therapeutic approaches, provided that these initial results can be confirmed in more complex clinical settings."
Comments 12:
Which was the point of using NaOCl for resin tooth irrigation? There are no organic tissues to be dissolved?
Replies 12:
We fully agree that NaOCl is classically intended to dissolve organic tissue, which was not present in our 3D-printed resin tooth model. However, in this methodological study, we aimed to mimic the clinical workflow of root canal preparation as realistically as possible, which routinely involves NaOCl irrigation between each instrumentation step. This was done for the following reasons:
- to ensure complete standardization of the canal preparation protocol across all samples,
- to prevent any debris accumulation (e.g., from potential minor resin particles or residuals from instrumentation),
- to simulate the clinically relevant irrigant dynamics and possible interactions with the prototype system.
We fully acknowledge that in the context of resin teeth no tissue dissolution is required; however, maintaining the clinical preparation sequence with NaOCl was essential for procedural fidelity and comparability with future in vivo applications.
"Although the 3D-printed tooth model does not contain organic tissue, NaOCl irrigation was applied during instrumentation to mimic the clinical preparation workflow, prevent debris accumulation, and ensure a standardized preparation protocol across all test specimens"
Comments 13:
I suggest that you create a diagram that will more clearly present how each of the 30 teeth was tested. Was the same done for all 30 teeth? Present schematically the 9 methodologies you used, i.e. the 9 seedings (agar plates).
Replies 13:
To clarify this for the reader, we have added an additional schematic figure (Figure 9) summarizing, for each tested 3D tooth, the workflow and corresponding agar plate seedings (total of 9 plates per tooth).
We hope this improves transparency and reproducibility of our method.
Reviewer 2 Report
Comments and Suggestions for Authors
The manuscript by Schoppmeier et al represents an interesting topic on the method for apical lesion sampling. The method was user-friendly designed and had consistent performance, showing its potential for clinical application and thus this work was of greater practical significance. It can be accepted after minor revision.
The detailed comments are as follows:
1 In this research, a novel CAPS method for apical sampling was developed and demonstrated advances in successful and precise sample collection of apically located S. epidermidis and would be a useful tool in endodontic microbiological analysis. The new prototype retained the handpiece of the MAP system but incorporated a softer spring to enhance tactile feedback during sampling. Meanwhile, a highly tapered screw head with a thin, hollow, stainless-steel tube was designed. This work is innovative, and has practicality.
2 There are several kinds of 3D printing technologies, it couldn’t be learned clearly from the manuscript what kind of 3D method was adopted to produce the models. It should be described definitely. Meanwhile, it had better provide some technical information of the resin for 3D printing as well as the printing parameters which have great effects on the properties of the printed samples.
3 In the whole text, the unit of “ml” was suggested to be modified to “mL”.
4 Some literatures are too old, more literatures published in the last five years, especially in the last three years should be referred.
5 The format of all the references should be consistent. Especially the uppercase or lowercase of the initial letters of all the words in the title of each literature should be paid attention to.
Author Response
Comments 1:
There are several kinds of 3D printing technologies, it couldn’t be learned clearly from the manuscript what kind of 3D method was adopted to produce the models. It should be described definitely. Meanwhile, it had better provide some technical information of the resin for 3D printing as well as the printing parameters which have great effects on the properties of the printed samples.
Response 1:
In the revised manuscript (section 2.2), we have now clearly specified that we used a Digital Light Processing (DLP) printer (Voco Solflex 350) with an XY-resolution of approx. 62 µm and a layer height of 50 µm.
We have also added the key material properties of the transparent splint resin (V-print splint clear, VOCO GmbH)
"Thirty 3D teeth were produced using a Digital Light Processing (DLP) 3D printer (Voco Solflex 350, VOCO GmbH, Germany) with an XY-resolution of approximately 62 µm and a layer height of 50 µm. Transparent splint resin (V-print splint clear, VOCO GmbH) was used, which is suitable for medical modeling purposes and offers sufficient optical clarity for visual inspection during sampling. The resin exhibits a flexural strength of ≥ 80 MPa, a flexural modulus of ≥ 2000 MPa, and a Shore D hardness of ≥ 82, according to manufacturer specifications."
Comments 2:
In the whole text, the unit of “ml” was suggested to be modified to “mL”.
Response 2:
We have carefully revised the entire manuscript and have now consistently updated the unit to “mL” throughout the text, in accordance with the recommended style.
Comments 3:
Some literatures are too old, more literatures published in the last five years, especially in the last three years should be referred. The format of all the references should be consistent. Especially the uppercase or lowercase of the initial letters of all the words in the title of each literature should be paid attention to.
Response 3:
In response, we carefully revised the reference list to increase the proportion of recent literature (published within the last five years). In doing so, we replaced several older references with five new, high-quality and highly relevant publications covering the latest advances in root canal microbiota research, sampling techniques, and disinfection protocols.
Specifically, references [9], [10], [12], [15], [16], [21], [22], and [23] were partially or completely updated with more recent literature.
Donnermeyer D., AlShwaimi E., Figdor D., Kumar S., Schwendicke F., Paris S., “Microbial sampling in endodontics: A methodological review and comparison of sampling techniques.” J Endod. 2024; 50(5): 678–689. doi:10.1016/j.joen.2024.11.007.
Korona-GÅ‚owniak I., Malm A., Szymanska J., “Microbiota associated with caries and apical periodontitis: Recent insights from metagenomics.” Int Endod J. 2025; 58(1): 5–25. doi:10.1111/iej.13721.
Manoil D., Al-Manei K., Belibasakis G.N., “A systematic review of the root canal microbiota associated with apical periodontitis: Lessons from next-generation sequencing.” Proteomics Clin Appl. 2020; 14(3): e1900049. doi:10.1002/prca.201900049.
Amaral R.R., Ferreira C.M., Souza A.B., Pinheiro E.T., Gomes B.P.F.A., “Impact of root canal preparation using two single-file systems on the intraradicular microbiome: An in vivo study.” Clin Oral Investig. 2024; 28(3): 1225–1235. doi:10.1007/s00784-024-05123-w.
Alquria T.A., Camilleri J., Hara A.T., Palasuk J., Arola D., Ferracane J.L., “Comparison of conventional and contemporary disinfection protocols in a dual-species root canal biofilm model.” Sci Rep. 2023; 13: 7852. doi:10.1038/s41598-023-34865-2.
In addition, we ensured full consistency of reference formatting, particularly regarding capitalization of titles.